# Association between mean platelet volume and obstructive sleep apnea-hypopnea syndrome: A systemic review and meta-analysis

Jun Zeng[1,2,3☉], Jie He[1,2,3☉]*, MeiFeng Chen[1,2,3], Jia Li[1,2,3]

**1** Clinical Medical College of Chengdu Medical College, Chengdu, Sichuan, China, **2** Department of Pulmonary and Critical Care Medicine, The First Affiliated Hospital of Chengdu Medical College, Chengdu, Sichuan, China, **3** Key Laboratory of Geriatric Respiratory Diseases of Sichuan Higher Education Institutes, Chengdu, Sichuan, China

☉ These authors contributed equally to this work.
* 13540246974@163.com.

## Abstract

### Background

Despite polysomnography (PSG) being acknowledged being considered the gold standard for diagnosing obstructive sleep apnea-hypopnea syndrome (OSAHS), researchers have been seeking a biomarker that is less invasive, more practical in detection, and cost-effective for diagnosing and assessing the severity of the disease. To address this concern, the values of mean platelet volume (MPV) between patients with OSAHS and healthy controls were compared, and the relationship between MPV and multiple sleep monitoring parameters was analyzed in this study.

### Methods

A comprehensive search was conducted across medical databases, including PubMed, Web of Science, EMBASE, CNKI, and Wanfang, up until August 2, 2023, to identify published articles related to OSAHS. This study reviewed the literature regarding the values of MPV in individuals with OSAHS and control groups, the Pearson/Spearman correlation coefficients between MPV and sleep monitoring parameters, and the odds ratios (OR) of MPV concerning the occurrence of cardiovascular diseases (CVDs) in patients with OSAHS. Meta-analyses were performed using standardized mean difference (SMD), Fisher's z values correlation coefficients (ZCOR) and odds ratio (OR) as effect variables. A fixed-effect model was used if the heterogeneity was not significant ($I^2 < 50\%$); otherwise, a random-effect model was applied. We will also combine the treatment effect estimates of individual trials using fixed-effect and random-effects models. Statistical analysis was carried out by employing STATA 11.0 and R 4.1.3.

### Results

In total, 31 articles were selected for the final analysis. The study involved 3604 patients and 1165 control individuals. The MPV in the OSAHS group was considerably elevated in comparison to the healthy controls (SMD = 0.37, 95%CI = 0.21–0.53, $P < 0.001$), particularly

**Data Availability Statement:** All relevant data are within the paper.

**Funding:** Priming Scientific Research Foundation for the Introduced Talents of The First Affiliated

Hospital of Chengdu Medical College(CYFY-GQ59) The funders had no role in study design, data collection and analysis, decision to publish, or preparation of the manuscript.

among individuals with severe OSAHS (SMD = 0.57, 95%CI = 0.23–0.90, $P$ = 0.001). Subgroup analysis based on ethnicity, mean body mass index (BMI), and study design type also revealed a considerably higher MPV in the OSAHS category in comparison to the healthy controls. Furthermore, MPV showed correlations with various sleep monitoring parameters. The elevation of MPV may be one of the risk factors for CVDs in individuals with OSAHS (adjusted OR = 1.72, 95%CI = 1.08–2.73, $P$ = 0.022).

## Conclusion

MPV is a relatively simple, cost-effective, and practical indicator of the severity of OSAHS, with its values being linked to the risk of CVDs in individuals with OSAHS.

## Introduction

Obstructive sleep apnea-hypopnea syndrome (OSAHS) is characterized by recurrent episodes of complete (apnea) or partial (hypopnea) blockage of the upper airway while sleeping, which results in a repetitive pattern of alternative hypoxia and reoxygenation [1]. The prevalence of OSA is increasing as the global obesity epidemic worsens, and OSA currently affects 9% of women and 24% of men [2]. In the United States, around 26% of individuals aged 30–70 years are estimated to have OSAHS [3], although this data may be underestimated.

At present, polysomnography (PSG) is widely regarded as the gold standard for clinically diagnosing OSAHS [4]. However, there are certain situations where obtaining PSG results can be challenging. One reason is the difficulty in setting up PSG monitoring devices and sleep monitoring rooms in primary healthcare facilities. Another reason is that for children, PSG examinations require them to sleep throughout the night, and the results may not be promptly available for feedback.Therefore, identifying a simple, easily obtainable outpatient indicator that aligns with PSG results would aid cliniciansin the initial evaluation of the condition and the development of subsequent treatment plans. It had been shown that Endocan and YKL-40 levels in serum, and IL-6 and Vimentin levels in plasma revealed the most promising candidates for OSA diagnosis [5, 6]. Fiedorczuk et al. found that a total of 14 sets of candidate biomarkers displayed differences in levels or concentrations in OSAHS patients compared to non-OSA controls, and decreased after OSA treatment: IL-6, leptin, IL-8, IGF-1, etc [7]. The relevant parameters in blood analysis panels are relatively inexpensive and easily obtainable, while also proven to have significant clinical value [8]. Mean platelet volume (MPV), as a routine parameter in platelet morphology, has increasingly been studied as a biomarker in clinical medicine [9, 10]. MPV reflects platelet activation and is associated with various inflammatory conditions [11]. Platelets with a larger size show an elevated quantity of granules and thromboxane A2, exhibiting heightened expression of glycoprotein Ib and IIb/IIIa receptors. Consequently, these platelets demonstrate rapid and more efficient aggregation of collagen, thereby contributing to an elevated risk of thrombotic events [12–14]. Prior evidence has highlighted that OSAHS causes platelet activation [15]. Additionally, the activation of platelets during sleep may increase in individuals with OSAHS, leading to a higher incidence of CVDs [16–19]. In individuals with OSAHS, a substantial increase in MPV has been demonstrated to correlate positively with the severity of the disease. The variations in MPV values in individuals with OSAHS may be attributed to the activation of pro-inflammatory factors induced by intermittent hypoxia [20, 21].

Nevertheless, it is still debated whether people with OSAHS have an increased MPV compared with people without OSAHS. The relationship between the increase of MVP and

OSAHS is still unclear.Therefore, a systematic review and meta-analysis are executed further to investigate the difference in MPV values for OSAHS and the potential association between MPV values and OSAHS, aiming to determine the clinical value of this parameter in assessing the severity of OSAHS.

## Methods

This systematic review protocol has been registered on PROSPERO (https://www.crd.york.ac. uk/PROSPERO/display_record.php?RecordID=451081) with number PROSPERO CRD42023451081. This research adhered to the guidelines outlined in the Preferred Reporting Items for Systematic Reviews and Meta-Analysis (PRISMA) statement, which involves a systematic assessment and meta-analysis [22].

### Data sources and search strategy

A comprehensive search was conducted across medical databases, namely PubMed, Web of Science, EMBASE, CNKI, and Wanfang. The search spanned from the inception of each database up to August 2, 2023. The search strategy included the following keywords: ("MPV" or "mean platelet volume") and ("obstructive sleep apnea" or "OSA" or "sleep apnea" or "obstructive sleep apnea-hypopnea syndrome" or "OSAS" or "obstructive sleep apnea syndrome" or "OSAHS"). Moreover, the reference lists of pertinent articles were reviewed to identify any potentially overlooked studies. The exact search terms of each database were shown in S1 Table.

### Inclusion and exclusion criteria

We included studies reporting human-only original research (randomized-controlled trials, prospective or retrospective comparatives cohorts, and case-control studies).

The inclusion criteria for this meta-analysis were as follows: (I) Patients: Adults or children with OSAHS. The individuals were required to meet the diagnostic criteria for OSAHS as determined by PSG (adults: AHI ≥ 5/h; children: AHI ≥ 1/h) [23]. The severity of OSAHS was categorized based on conventional definitions (adults: AHI < 5, normal; AHI 5–14, mild OSAHS; AHI 15–29, moderate OSAHS; and AHI ≥ 30, severe OSAHS; children: AHI < 1, normal; AHI 1–4, mild OSAHS; AHI 5–9, moderate OSAHS; and AHI ≥ 10, severe OSAHS) [24, 25]; (II) Outcome: MPV values; (III) Language: Chinese and English languages; (IV) Data: All data were based on mean and standard deviation or median and range. The exclusion criteria were as follows: studies with unavailable data, animal experiments, reviews, case reports, or letters were excluded from the analysis.

### Quality assessment of the literature

The methodological quality of the studies included in the analysis was evaluated by employing the cross-sectional study assessment tool recommended by the Agency for Healthcare Research and Quality (AHRQ) [26]. Each item in the tool was scored as "0" if the answer was "No" or "Unclear," or "1" if the answer was "Yes". The quality of the articles was categorized as follows: low quality = 0–3, medium quality = 4–7, and high quality = 8–11. The Newcastle-Ottawa Scale (NOS) [27] was employed to evaluate cohort and case-control studies. It consisted of three aspects: study subjects (4 items, total score of 4), exposure or outcome (3 items, total score of 3), and comparability between exposed and non-exposed groups (1 item, total score of 2). Studies were considered high quality, medium quality, or low quality if they scored 7–9, 4–6, or 0–3, respectively.

## Data extraction

Two authors independently evaluated all studies retrieved from databases and bibliographies and determined the final inclusion of studies based on the aforementioned inclusion criteria. Data from each publication were entered into a standardized electronic spreadsheet format by the two authors, which included all relevant details, such as the first author's name, publication year, study design type, sample size, relevant exposures and interventions, control variables, statistical methods, MPV values, and correlation coefficients between MPV and OSAHS monitoring parameters. Any disagreements were resolved through consensus or by seeking guidance from a third-party reviewer.

## Statistical analysis

Statistical analysis was carried out by employing STATA 11.0(Stata Corporation, College Station, TX, United States) and R 4.1.3(R Project for Statistical Computing, http://www.r-project.org/). The meta-analysis was conducted using the method of residual maximum likelihood (REML).A meta-analysis was conducted using standardized mean difference (SMD) and 95% confidence interval (CI) to examine the differences in MPV values among individuals with OSAHS and healthy controls. The relationship between MPV and OSAHS monitoring parameters, such as AHI and ODI, was assessed using Pearson correlation coefficients. The Z-values were transformed utilizing Fisher's z-transformation, and Pearson's correlation coefficients were calculated. Some articles provided Spearman correlation coefficients, which were converted to Pearson correlation coefficients before being included in the meta-analysis. The conversion formula between Spearman and Pearson correlation coefficients is as follows:

$$r = 2 \sin \left( r_s \frac{\pi}{6} \right)$$

Wherein the respective Pearson's and Spearman's CORs are denoted by r and $r_s$. The correlation was classified as (r): 0–0.1 very low; 0.1–0.3 low; 0.3–0.5 moderate; 0.5–0.7 high; 0.7–0.9 very high; 0.9–1.0 almost perfect [28]. The R code is shown in S2 Table. The $I^2$ statistic was used to assess study heterogeneity.Statistical heterogeneity was detected using the Q statistic and $I^2>50\%$ indicated high heterogeneity [29, 30]. Fixed or random-effects models were used depending on the between-study heterogeneity (threshold at >50%). Moreover, We will estimate the treatment effect across studies using the fixed-effect or random-effects model, and use the fixed-effect model if it is reasonable to assume that studies are estimating the same under lying treatment effect.The effect size of each study was also analyzed by removing them in turn in sensitivity analyses. Possible differences between studies were explored by subgroup analyses and meta-regression. We performed the subgroup analyses to determine the efforts of ethnicity, disease severity, BMI and research style on the MPV values between OSAHS patents and control. We also introduced a meta-regression model with ethnicity, disease severity, BMI and research style as independent variables to explore possible heterogeneity. To investigate the presence of publication bias, Begg's adjusted rank correlation test and Egger's regression asymmetry test were used.

# Results

## Literature selection

Overall, 240 studies related to the topic were retrieved from the databases. Following the elimination of duplicate entries, 107 studies were screened. After the exclusion of 61 irrelevant articles based on the screening of abstracts and titles, the total number of studies was reduced to

46. Following inclusion and exclusion per the pre-defined criteria, 15 publications were excluded. Among these, three were review articles, three were letters to the editors, seven lacked available data, and two involved animal experiments. Finally, 31 articles (_((((xxx))))_) [17, 20, 21, 31–58] were included in the meta-analysis. The study involved 3604 patients and 1165 control individuals. Among these included studies, 23 articles (_((((xxx))))_)[17, 20, 21, 31–41, 44–47, 49, 50, 52, 55, 56] compared the differences in MPV between individuals with OSAHS and healthy controls. Additionally, 19 articles (_((((xxx))))_)[17, 20, 21, 32, 34–41, 44, 46, 47, 49, 50, 52, 56] reported the Pearson or Spearman correlation coefficients of MPV with various sleep monitoring parameters (AHI: n = 17, mean arterial oxygen saturation [mean SaO$_2$]: n = 4, ODI: n = 5, sleep efficiency: n = 2, SaO$_2$ <90%: n = 8, LSaO$_2$: n = 7). Further-more, three articles [20, 35, 43] provided adjusted ORs for the association between MPV and the occurrence of CVDs in patients with OSAHS (two articles focused on coronary heart disease and one article on atrial fibrillation). Fig 1 presents the literature screening process. The basic information on the outcomes of the selected studies is illustrated in Table 1. Based on the NOS and the AHRQ scale, the included studies were of relatively high quality, ranging between medium and high.

## Meta-analysis: Differences in MPV between OSAHS subjects and healthy controls

Among the 23 studies included in the meta-analysis, the I$^2$ heterogeneity index was 89%. Consequently, a random-effect model was applied to assess the reliability of the data. The results of the meta-analysis revealed that individuals with OSAHS had elevated MPV values compared to the healthy controls (SMD = 0.37, 95% CI = 0.21–0.53, $P$ < 0.001) (Fig 2).

## Subgroup analysis

Several included articles reported MPV values in individuals with OSAHS of different severities. Therefore, subgroup analyses were conducted based on mild, moderate, and severe OSAHS. After obtaining the demographic information of all individuals with OSAHS, we further carried out subgroup analyses based on ethnicity, body mass index (BMI), and study design type.

**Severity of disease.** Of note, most relevant studies reported the correlation between the severity of OSAHS and MPV compared to the controls. Hence, in the current meta-analysis, individuals were categorized into mild, moderate, and severe OSAHS based on the AHI. Thirteen studies provided data on MPV values in individuals with mild OSAHS. The obtained data suggested that no marked variance existed in MPV between patients with mild OSAHS and the control group (SMD = 0.23, 95% CI = -0.01–0.46, $P$ = 0.131). Thirteen studies provided data on MPV values in individuals with moderate OSAHS. The findings suggested that individuals with moderate OSAHS had higher MPV values compared to the controls (SMD = 0.38, 95% CI = 0.05–0.71, $P$ = 0.026). Sixteen studies examined MPV in both the controls and individuals with severe OSAHS. Interestingly, the resulting data indicated that individuals with severe OSAHS had higher MPV values than the controls (SMD = 0.57, 95%CI = 0.23–0.90, $P$ = 0.001) (Fig 3).

**Ethnicity.** Subgroup analysis can be used to explain the reasons for heterogeneity. According to ethnicity, participants were categorized into Caucasian, Asian, and African populations. Among the Caucasian population, the MPV in the case group was elevated relative to the healthy controls (SMD = 0.33, 95% CI = 0.16–0.51, P = 0.003). In the Asian population, the MPV in the case group was elevated than the healthy controls (SMD = 0.55, 95% CI = 0.25–

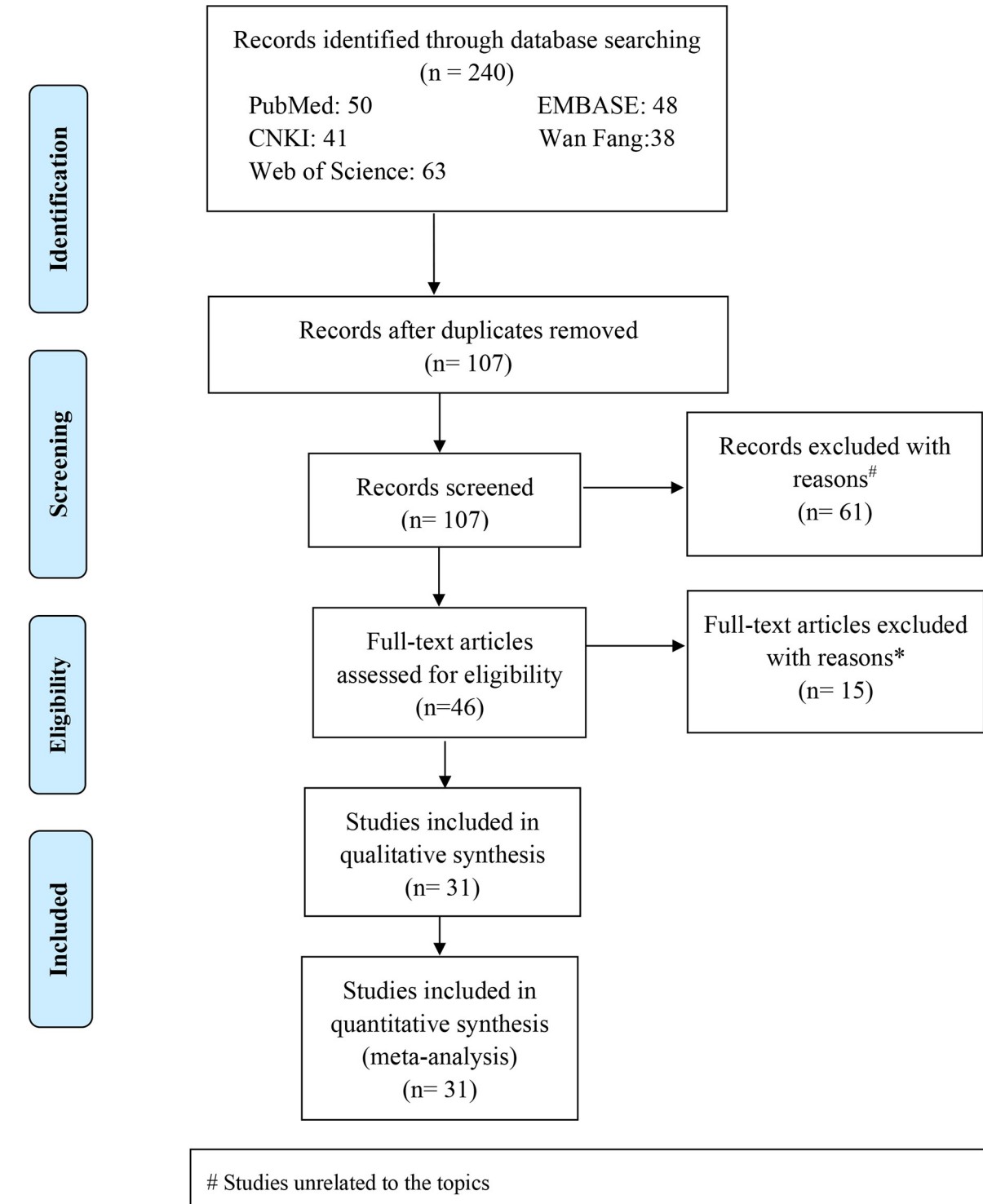

**Fig 1. Flow diagram of literature selecting process and results according to the preferred reporting items for the meta-analysis.**

**Table 1. Characteristics of included studies.**

| Author | Year | Country | Sex(Female/male) | | Age(year) | | BMI (kg/m²) | | Study design | Quality assessment | Evaluation index |
|---|---|---|---|---|---|---|---|---|---|---|---|
| | | | Case | Control | Case | Control | Case | Control | | | MPV value, Pearson/Spearman correlation, adjusted OR |
| Han SZ | 2006 | China | 0/20 | 0/20 | 45.95 ±9.16 | 46.25 ±8.62 | NA | NA | Case-control study | 5 | MPV value |
| Kanbay A(mild) | 2013 | Turkey | 7/13 | 13/22 | 55.6 ±11.6 | 51.2 ±12.6 | 31.4 ±5.2 | 29.2±8.4 | Cross-sectional study | 7 | MPV value, Pearson correlation coefficients, adjusted OR |
| Kanbay A(moderate) | 2013 | Turkey | 22/20 | 13/22 | 52.9 ±13.3 | 51.2 ±12.6 | 31.7 ±7.5 | 29.2±8.4 | Cross-sectional study | 7 | MPV value, Pearson correlation coefficients, adjusted OR |
| Kanbay A(severe) | 2013 | Turkey | 34/74 | 13/22 | 55.5 ±11.9 | 51.2 ±12.6 | 32.4 ±6.1 | 29.2±8.4 | Cross-sectional study | 7 | MPV value, Pearson correlation coefficients, adjusted OR |
| Karakas MS (mild) | 2013 | Turkey | NA | NA | 46.1±8.2 | 46.7±8.4 | 28.4 ±3.1 | 28.9±2.9 | Case-control study | 6 | MPV value, Spearman correlation coefficients |
| Karakas MS (moderate) | 2013 | Turkey | NA | NA | 48.3±7.6 | 46.7±8.4 | 28.7 ±2.7 | 28.9±2.9 | Case-control study | 6 | MPV value, Spearman correlation coefficients |
| Karakas MS (severe) | 2013 | Turkey | NA | NA | 47.3±7.7 | 46.7±8.4 | 29.2 ±2.9 | 28.9±2.9 | Case-control study | 6 | MPV value, Spearman correlation coefficients |
| Akyuz A | 2014 | Turkey | 13/39 | 9/21 | 48±11 | 44±11 | 32.7 ±4.9 | 30.0±7.1 | Cohort study | 6 | MPV value, Pearson/Spearman correlation coefficients |
| Gunbatar H | 2014 | Turkey | NA | NA | 50.8 ±11.7 | 41.3 ±11.0 | 32.9 ±5.0 | 27.3±4.5 | Cohort study | 7 | MPV value, Pearson correlation coefficients |
| Sokucu SN | 2014 | Turkey | 0/36 | 0/22 | 47.44 ±11.68 | 40.76 ±11.62 | 33.10 ±4.35 | 28.68 ±6.09 | Case-control study | 7 | MPV value, Pearson correlation coefficients |
| Zheng XS(mild to moderate) | 2015 | China | 12/20 | 11/19 | 41.2 ±12.1 | 42.6 ±11.3 | NA | NA | Case-control study | 6 | MPV value |
| Zheng XS(severe) | 2015 | China | 15/23 | 11/19 | 39.7 ±10.5 | 42.6 ±11.3 | NA | NA | Case-control study | 6 | MPV value |
| Bulbul Y | 2016 | Turkey | 68/105 | 25/18 | 53.9 ±10.8 | 42.3 ±10.5 | 34.4 ±7.3 | 31.1±6.3 | Case-control study | 8 | MPV value, Pearson correlation coefficients |
| Sarioglu N(mild) | 2017 | Barcelona | 10/30 | 15/31 | 44.8 ±10.0 | 41.7±9.0 | 29.6 ±5.5 | 29.6±5.4 | Cohort study | 8 | MPV value, Pearson correlation coefficients |
| Sarioglu N (moderate) | 2017 | Barcelona | 8/24 | 15/31 | 45.9±8.3 | 41.7±9.0 | 29.6 ±4.8 | 29.6±5.4 | Cohort study | 8 | MPV value, Pearson correlation coefficients |
| Sarioglu N(severe) | 2017 | Barcelona | 8/32 | 15/31 | 46.6 ±10.3 | 41.7±9.0 | 32.9 ±3.6 | 29.6±5.4 | Cohort study | 8 | MPV value, Pearson correlation coefficients |
| Yilmaz Avci A(mild) | 2017 | Turkey | 30/39 | 37/37 | 53±12 | 44±13 | 31±6 | 28±5 | Cohort study | 7 | MPV value, Spearman correlation coefficients |
| Yilmaz Avci A (moderate) | 2017 | Turkey | 16/39 | 37/37 | 55±12 | 44±13 | 32±4 | 28±5 | Cohort study | 7 | MPV value, Spearman correlation coefficients |
| Yilmaz Avci A (severe) | 2017 | Turkey | 19/80 | 37/37 | 57±14 | 44±13 | 33±5 | 28±5 | Cohort study | 7 | MPV value, Spearman correlation coefficients |
| Archontogeorgis K | 2018 | Greece | 90/234 | 22/60 | 53.7 ±12.5 | 47.2 ±13.3 | 35.7 ±7.8 | 31.3±7.9 | Cohort study | 6 | MPV value, Pearson correlation coefficients |
| Kivanc T(mild) | 2018 | Turkey | 9/44 | 17/35 | 43±10 | 41±12 | 29±4.5 | 29±4.3 | Cohort study | 7 | MPV value, Pearson/Spearman correlation coefficients |
| Kivanc T(moderate) | 2018 | Turkey | 18/47 | 17/35 | 50±11 | 41±12 | 31.3 ±4.3 | 29±4.3 | Cohort study | 7 | MPV value, Pearson/Spearman correlation coefficients |

*(Continued)*

**Table 1.** (*Continued*)

| Author | Year | Country | Sex(Female/male) | | Age(year) | | BMI (kg/m²) | | Study design | Quality assessment | Evaluation index |
|---|---|---|---|---|---|---|---|---|---|---|---|
| | | | Case | Control | Case | Control | Case | Control | | | **MPV value, Pearson/Spearman correlation, adjusted OR** |
| Kivanc T(severe) | 2018 | Turkey | 29/130 | 17/35 | 51±11 | 41±12 | 34±6 | 29±4.3 | Cohort study | 7 | MPV value, Pearson/Spearman correlation coefficients |
| Ulusoy B | 2020 | Turkey | 7/29 | 14/16 | 47.0 ±20.8 | 41.0 ±13.5 | 31.48 ±8.51 | 24.35 ±2.25 | Cohort study | 6 | MPV value |
| Zorlu D(mild) | 2021 | Turkey | 20/21 | 28/26 | 53.4 ±11.0 | 53.1 ±11.2 | NA | NA | Cohort study | 7 | MPV value, Spearman correlation coefficients |
| Zorlu D(moderate) | 2021 | Turkey | 22/32 | 28/26 | 54.5 ±10.3 | 53.1 ±11.2 | NA | NA | Cohort study | 7 | MPV value, Spearman correlation coefficients |
| Zorlu D(severe) | 2021 | Turkey | 16/42 | 28/26 | 55.2 ±14.5 | 53.1 ±11.2 | NA | NA | Cohort study | 7 | MPV value, Spearman correlation coefficients |
| Zota IM (mild) | 2022 | Romania | 13/20 | 15/16 | 54.06 ±15.37 | 49.55 ±14.01 | 32.34 ±5.44 | 32.11 ±5.16 | Cohort study | 7 | MPV value, Pearson correlation coefficients |
| Zota IM (moderate) | 2022 | Romania | 6/16 | 15/16 | 57.68 ±9.18 | 49.55 ±14.01 | 32.65 ±6.16 | 32.11 ±5.16 | Cohort study | 7 | MPV value, Pearson correlation coefficients |
| Zota IM (severe) | 2022 | Romania | 11/37 | 15/16 | 58.49 ±9.49 | 49.55 ±14.01 | 35.41 ±5.63 | 32.11 ±5.16 | Cohort study | 7 | MPV value, Pearson correlation coefficients |
| Kallel S(mild) | 2023 | Tunisia | 37/16 | 21/5 | 50.73 ±11.72 | 49.61 ±11.56 | 30.57 ±5.69 | 29.12 ±5.68 | Cohort study | 8 | MPV value, Spearman correlation coefficients, adjusted OR |
| Kallel S(moderate) | 2023 | Tunisia | 22/16 | 21/5 | 52.92 ±13.45 | 49.61 ±11.56 | 31.68 ±5.91 | 29.12 ±5.68 | Cohort study | 8 | MPV value, Spearman correlation coefficients, adjusted OR |
| Kallel S(severe) | 2023 | Tunisia | 32/26 | 21/5 | 53.18 ±7.97 | 49.61 ±11.56 | 32.85 ±5.78 | 29.12 ±5.68 | Cohort study | 8 | MPV value, Spearman correlation coefficients, adjusted OR |
| Varol E(mild to moderate) | 2010 | Turkey | 20/22 | 10/14 | 50.1±9.3 | 45.6 ±13.9 | 29.0 ±4.1 | 28.2±5.0 | Case-control study | 5 | MPV value, Pearson correlation coefficients |
| Varol E(severe) | 2010 | Turkey | 8/21 | 10/14 | 49.6 ±10.2 | 45.6 ±13.9 | 31.5 ±4.0 | 28.2±5.0 | Case-control study | 5 | MPV value, Pearson correlation coefficients |
| Varol E | 2011 | Turkey | 10/21 | 11/14 | 53.8±9.2 | 49.6±8.5 | 32.5 ±3.3 | 30.9±2.9 | Case-control study | 6 | MPV value |
| Nena E(mild) | 2012 | Greece | NA | NA | 53.4 ±12.5 | NA | 35±7.2 | NA | Cohort study | 6 | MPV value, Pearson correlation coefficients |
| Nena E(moderate) | 2012 | Greece | NA | NA | 53.4 ±12.5 | NA | 35±7.2 | NA | Cohort study | 6 | MPV value, Pearson correlation coefficients |
| Nena E(severe) | 2012 | Greece | NA | NA | 53.4 ±12.5 | NA | 35±7.2 | NA | Cohort study | 6 | MPV value, Pearson correlation coefficients |
| Kurt OK (mild) | 2013 | Turkey | 4/11 | 9/11 | 51.7±8.9 | 46.3 ±13.1 | 28.4 ±3.3 | 29.4±4.9 | Cohort study | 6 | Spearman correlation coefficients |
| Kurt OK (moderate) | 2013 | Turkey | 11/15 | 9/11 | 53.9 ±12.4 | 46.3 ±13.1 | 31.7 ±4.8 | 29.4±4.9 | Cohort study | 6 | Spearman correlation coefficients |
| Kurt OK (severe) | 2014 | Turkey | 12/25 | 9/11 | 58.1 ±10.9 | 46.3 ±13.1 | 33.2 ±5.7 | 29.4±4.9 | Cohort study | 6 | Spearman correlation coefficients |
| Sokucu SN (mild) | 2014 | Turkey | 8/30 | 15/15 | 43.50 ±12.15 | 38.43 ±12.79 | 29.10 ±4.51 | 26.91 ±4.61 | Cohort study | 7 | MPV value, Pearson correlation coefficients |
| Sokucu SN (moderate) | 2014 | Turkey | 8/33 | 15/15 | 47.27 ±10.95 | 38.43 ±12.79 | 30.01 ±4.52 | 26.91 ±4.61 | Cohort study | 7 | MPV value, Pearson correlation coefficients |
| Sokucu SN (severe) | 2014 | Turkey | 15/76 | 15/15 | 45.64 ±10.15 | 38.43 ±12.79 | 31.75 ±4.53 | 26.91 ±4.61 | Cohort study | 7 | MPV value, Pearson correlation coefficients |

(*Continued*)

**Table 1.** (Continued)

| Author | Year | Country | Sex(Female/male) | | Age(year) | | BMI (kg/m$^2$) | | Study design | Quality assessment | Evaluation index |
|---|---|---|---|---|---|---|---|---|---|---|---|
| | | | Case | Control | Case | Control | Case | Control | | | **MPV value, Pearson/ Spearman correlation, adjusted OR** |
| Koseoglu HI (mild) | 2015 | Turkey | 35/58 | 34/23 | 51.1±8.6 | 43.5 ±11.2 | 30.2 ±4.8 | 29±4.8 | Cohort study | 7 | MPV value, Pearson correlation coefficients |
| Koseoglu HI (moderate) | 2015 | Turkey | 20/62 | 34/23 | 51.1 ±10.9 | 43.5 ±11.2 | 32.4 ±6.4 | 29±4.8 | Cohort study | 7 | MPV value, Pearson correlation coefficients |
| Koseoglu HI (severe) | 2015 | Turkey | 53/ 139 | 34/23 | 51.6 ±10.6 | 43.5 ±11.2 | 35.3 ±7.2 | 29±4.8 | Cohort study | 7 | MPV value, Pearson correlation coefficients |
| Uygur F(mild) | 2016 | Turkey | 21/36 | 57/61 | 53.7 ±10.8 | 50.3 ±11.7 | 30.8 ±5.7 | 29.4±7.8 | Cohort study | 7 | MPV value, Spearman correlation coefficients |
| Uygur F(moderate) | 2016 | Turkey | 23/30 | 57/61 | 51.8 ±12.1 | 50.3 ±11.7 | 31.6 ±8.1 | 29.4±7.8 | Cohort study | 7 | MPV value, Spearman correlation coefficients |
| Uygur F(severe) | 2016 | Turkey | 22/39 | 57/61 | 54.5 ±12.7 | 50.3 ±11.7 | 32.1 ±7.1 | 29.4±7.8 | Cohort study | 7 | MPV value, Spearman correlation coefficients |
| Akyuz A | 2015 | Turkey | 11/29 | 14/36 | 63.8±7.0 | 61.9±5.8 | 28.5±4 | 27.5±4 | Cohort study | 6 | MPV value, Spearman correlation coefficients, adjusted OR |
| Akyol S(mild) | 2015 | Turkey | 19/43 | NA | 45.5 ±11.2 | NA | 30.0 ±5.5 | NA | Cohort study | 7 | MPV value, Pearson correlation coefficients |
| Akyol S(moderate) | 2015 | Turkey | 13/48 | NA | 46.2 ±11.7 | NA | 30.1 ±3.6 | NA | Cohort study | 7 | MPV value, Pearson correlation coefficients |
| Akyol S(severe) | 2015 | Turkey | 14/57 | NA | 46.5 ±10.8 | NA | 32.1 ±3.6 | NA | Cohort study | 7 | MPV value, Pearson correlation coefficients |
| Zicari AM | 2016 | Italy | 16/29 | 30/40 | 7.62 ±3.09 | 9.04 ±3.91 | 16.78 ±0.40 | 17.56 ±4.8 | Case-control study | 7 | MPV value, Pearson correlation coefficients |
| Gong W | 2018 | China | 6/62 | 9/50 | 55.9 ±10.5 | 54.5±9.2 | 27.4 ±3.0 | 25.9±2.7 | Cross-sectional study | 6 | Spearman correlation coefficients |
| Barcelo A | 2019 | Spain | 37/50 | 29/36 | 8.3±3.5 | 8.6±3.5 | 24.7 ±9.1 | 23.3±8.5 | Cross-sectional study | 6 | MPV value, Spearman correlation coefficients |
| Chen M | 2022 | China | 50/ 229 | 18/54 | 49.83 ±13.13 | 55.79 ±14.18 | 27.8 ±2.9 | 25.2±3.2 | Cross-sectional study | 6 | MPV value, Spearman correlation coefficients |
| Zeng GH | 2022 | China | NA | NA | 5.82 ±2.21 | 5.82 ±2.21 | NA | NA | Case-control study | 6 | Pearson correlation coefficients |
| Wang Y(mild-moderate) | 2019 | China | 3/42 | NA | 41.9±1.6 | NA | 27.4 ±2.5 | NA | Case-control study | 6 | MPV value, Spearman correlation coefficients |
| Wang Y(severe) | 2019 | China | 6/54 | NA | 42.1±1.1 | NA | 26.2 ±2.2 | NA | Case-control study | 6 | MPV value, Spearman correlation coefficients |

MPV: Mean platelet volume, BMI: Body Mass Index, NA: not available

0.86, $P < 0.001$). Similarly, in the African population, the MPV in the case group was elevated than the healthy controls (SMD = 0.73, 95% CI = 0.22–1.25, $P = 0.005$) (Table 2).

**Mean BMI.** Given that several studies reported the BMI data of the individuals, a subgroup analysis was conducted based on whether the mean BMI exceeded 30. Thirty-five studies focused on OSAHS cases with a mean BMI greater than 30. The results of the pooled effect size showed that OSAHS cases with a mean BMI greater than 30 had higher MPV values in comparison to the controls (SMD = 0.44, 95% CI = 0.24–0.64, $P = 0.001$). Nine studies focused on OSAHS cases with a mean BMI of less than 30. The results of the pooled effect size

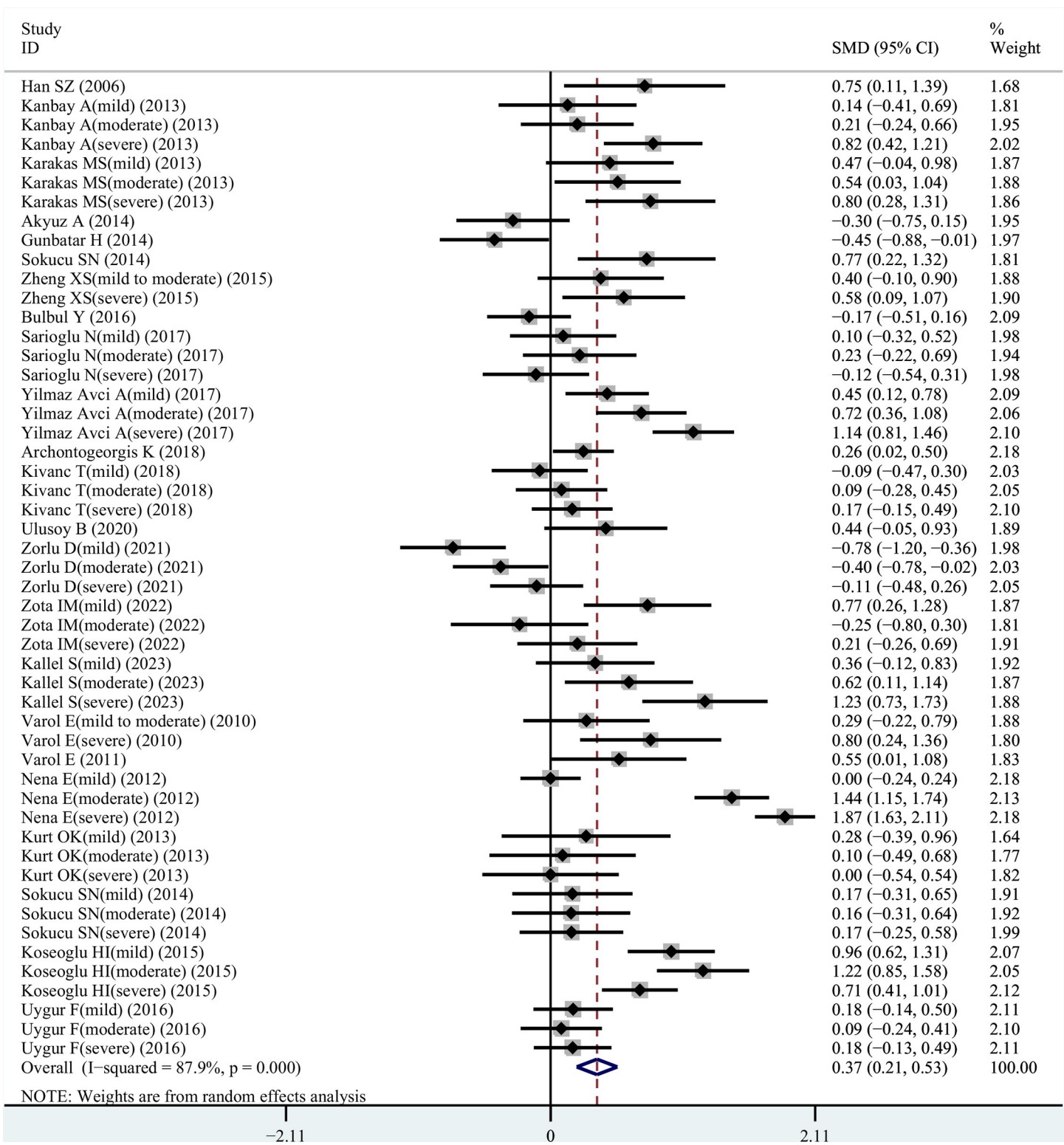

**Fig 2. Forest plot of SMD and its 95%CI for MPV levels in integral OSAHS patients group compared to the control group in meta-analysis.** SMD: standard mean difference; CI: confidence interval; OSAHS: obstructive sleep apnea-hypopnea syndrome. NOTE: Weights are from random effects analysis.

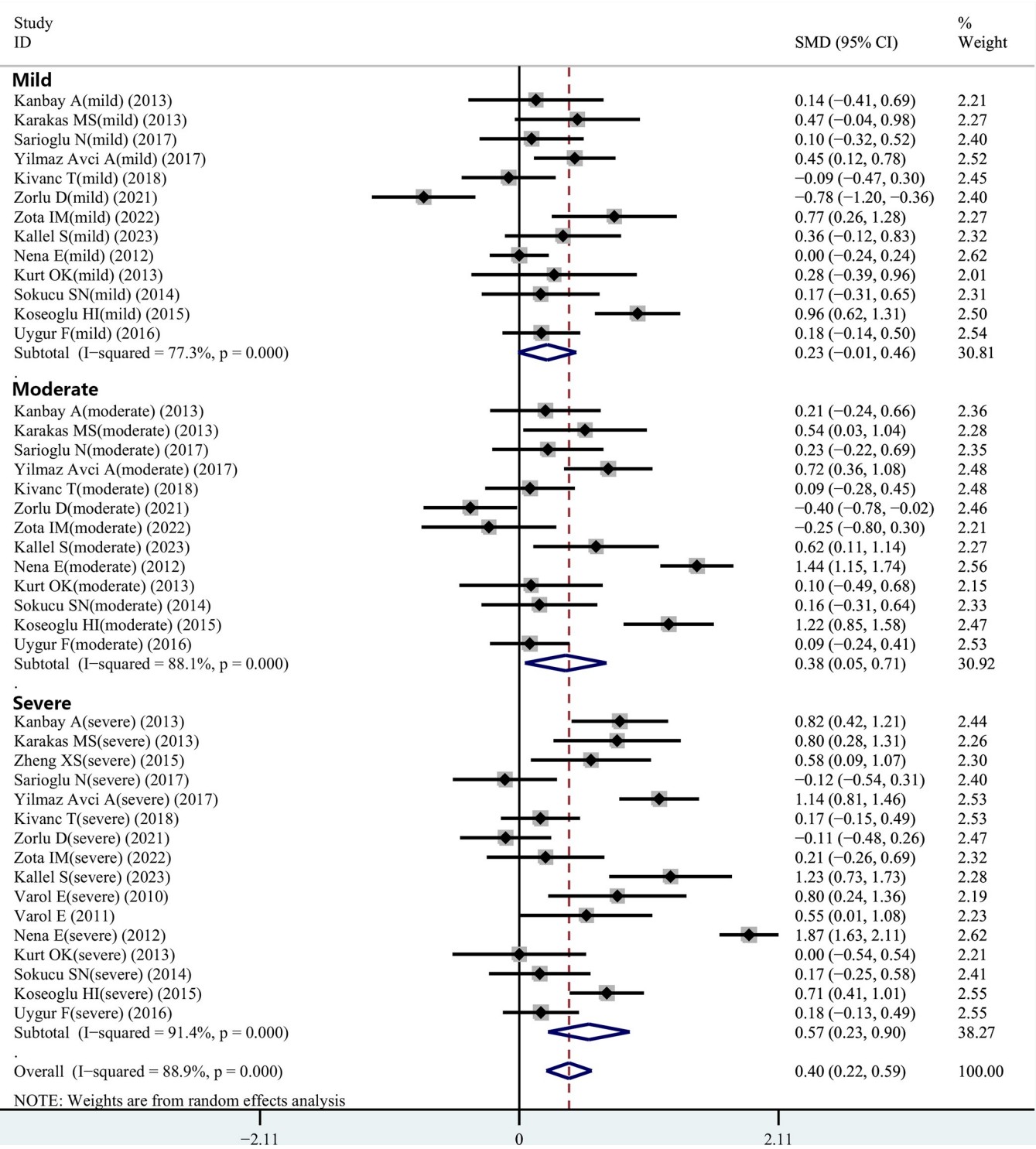

**Fig 3. Subgroup analysis depended on different severity of OSAHS patients compared to the controls was displayed in meta-analysis.** The means of expression are SMD and its 95%CI. SMD: standard mean difference; CI: confidence interval; OSAHS: obstructive sleep apnea-hypopnea syndrome. NOTE: Weights are from random effects analysis.

**Table 2. Subgroup analyses of the association between MPV and OSAHS in this meta-analysis.**

| Subgroup | No. of studies | SMD | 95%CI | P for heterogeneity | $I^2$(%) | P value between groups |
|---|---|---|---|---|---|---|
| Ethnicity | | | | | | |
| Asian | 3 | 0.55 | 0.25,0.86 | 0.694 | 0 | <0.001 |
| Caucasian | 45 | 0.33 | 0.16,0.51 | <0.001 | 89.1 | <0.001 |
| African | 3 | 0.73 | 0.22,1.25 | 0.04 | 68.9 | 0.005 |
| Disease severity | | | | | | |
| Mild | 13 | 0.23 | -0.01, -0.46 | <0.001 | 75.7 | 0.059 |
| Moderate | 13 | 0.38 | 0.05,0.71 | <0.001 | 88.1 | 0.026 |
| Severe | 16 | 0.57 | 0.23,0.90 | <0.001 | 91.4 | 0.001 |
| Mean BMI | | | | | | |
| >30 | 35 | 0.44 | 0.24,0.64 | <0.001 | 89.9 | <0.001 |
| <30 | 9 | 0.28 | 0.1,0.46 | 0.282 | 18.1 | 0.002 |
| Research style | | | | | | |
| Case-control study | 11 | 0.49 | 0.28,0.70 | 0.035 | 48.6 | <0.001 |
| Cross-sectional study | 3 | 0.41 | -0.03, -0.86 | 0.058 | 64.8 | 0.07 |
| Cohort study | 37 | 0.33 | 0.12,0.53 | <0.001 | 90.7 | 0.002 |

highlighted that OSAHS cases with a mean BMI less than 30 had higher MPV values in comparison to the controls (SMD = 0.28, 95% CI = 0.10–0.46, *P* = 0.002) (Table 2).

**Study design types.** Three studies were cross-sectional studies. The meta-analysis results showed no statistically significant difference in MPV between individuals with OSAHS and the controls (SMD = 0.414, 95% CI = -0.03–0.86, *P* = 0.070). Eleven studies were case-control studies, and the meta-analysis data revealed that individuals with OSAHS had higher MPV compared to the controls (SMD = 0.487, 95% CI = 0.28–0.70, *P* < 0.001). Thirty-seven studies were cohort studies. The meta-analysis results revealed that individuals with OSAHS had higher MPV compared to the controls (SMD = 0.33, 95% CI = 0.123–0.53, *P* = 0.017) (Table 2).

## Sensitivity analysis and meta-regression

While comparing MPV differences between the patients and the controls, high heterogeneity was observed in the meta-analysis results ($I^2$ = 89%). Therefore, a meta-regression was conducted to determine the source of this heterogeneity. The data acquired demonstrated that the *P*-values for covariates such as ethnicity, disease severity, mean BMI and study design types were 0.713, 0.096, 0.308, and 0.229, respectively. This information indicates that these factors mentioned earlier did not have a significant impact on the heterogeneity. After systematically excluding each study, any research that significantly influenced the final results could not be identified. The sensitivity analysis confirmed the stability of this meta-analysis (Fig 4A).

## Publication bias

A funnel plot was used to evaluate publication bias regarding the size of MPV differences between individuals with OSAHS and the controls in the included studies. The results indicated no publication bias (Fig 4B).

## Descriptive analysis of differences in MPV between children with OSAHS and controls

The study by Zicari *et al*. [48] included 22 children with OSAHS, 45 children with simple snoring, and 70 healthy children. They compared the differences in MPV across three groups and

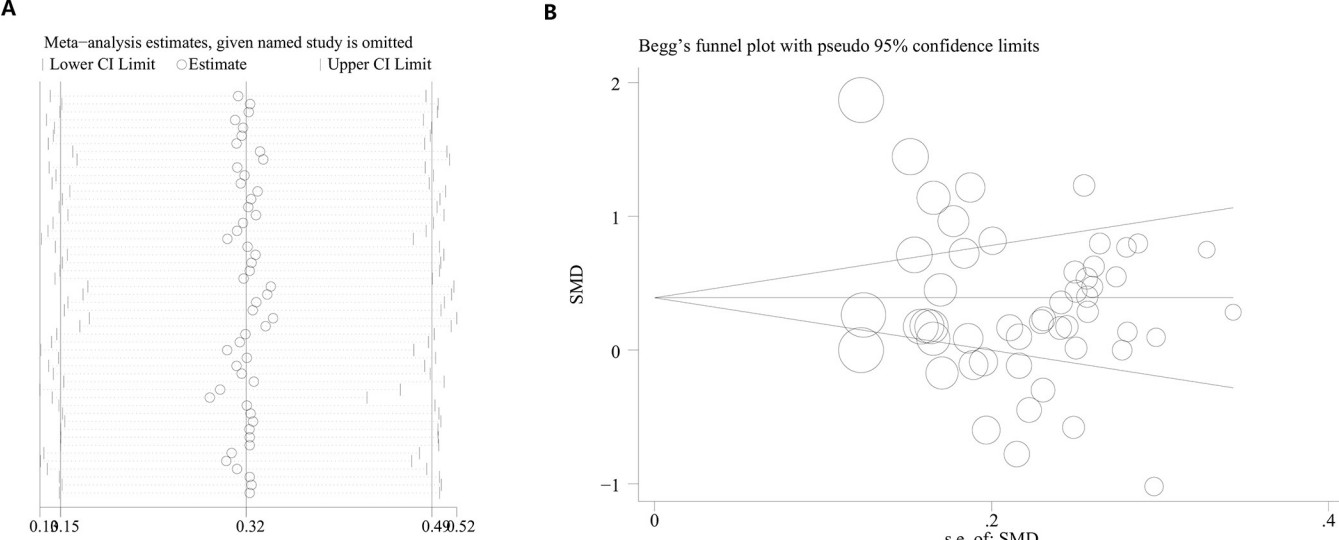

**Fig 4. Sensitivity analysis and publication bias.** A: Sensitivity analysis of the influence of subsequent testing on the overall findings by systematically excluding specific studies in the meta-analysis. B: Funnel plot was employed to evaluate the publication bias among those included literatures about the association between MPV and OSAHS. OSAHS: obstructive sleep apnea-hypopnea syndrome.

found that children with OSAHS had considerably elevated MPV values compared to healthy children (8.48 ± 0.93 fL *vs.* 7.57 ± 0.56 fL, $P < 0.05$). Additionally, children with OSAHS exhibited significantly higher MPV values compared to children with simple snoring (8.48 ± 0.93 fL *vs.* 8.09 ± 0.94 fL, $P < 0.05$).

## Meta-analysis: Relationship between MPV and sleep monitoring parameters

**Correlation between MPV and AHI.**   Seventeen studies have documented the correlation between MPV and AHI in the included population. After pooling the Pearson correlation coefficients, the meta-analysis results indicated a weak correlation between MPV and AHI (ZCOR = 0.29, 95% CI = 0.16–0.42, $P < 0.001$) (Fig 5A).

**Correlation between MPV and mean $SaO_2$.**   Four studies have delineated the correlation between MPV and mean $SaO_2$ in the study population. After pooling the Pearson correlation coefficients, the meta-analysis results showed no correlation between MPV and mean $SaO_2$ (ZCOR = -0.04, 95% CI = -0.20–0.13, $P = 0.656$) (Fig 5B).

**Correlation between MPV and ODI.**   The correlation between MPV and ODI in the included population has been examined in five studies. After pooling the Pearson correlation coefficients, the meta-analysis results showed a weak correlation between MPV and ODI (ZCOR = 0.25, 95% CI = 0.05–0.44, $P = 0.014$) (Fig 5C).

**Correlation between MPV and sleep efficiency.**   Two studies have yielded results regarding the correlation between MPV and sleep efficiency in the studied population. After pooling the Pearson correlation coefficients, the meta-analysis results indicated a weak correlation between MPV and sleep efficiency (ZCOR = 0.15, 95% CI = 0.01–0.29, $P = 0.030$) (Fig 5D).

**Correlation between MPV and percentage of total sleep time spent with $SaO_2 < 90\%$ (TS90%).**   In the included population, eight studies have clarified the association between MPV and TS90%. The meta-analysis, which pooled the Pearson correlation coefficients, showed a modest correlation between MPV and TS90% in the findings (ZCOR = 0.25, 95% CI = 0.17–0.34, $P < 0.001$) (Fig 5E).

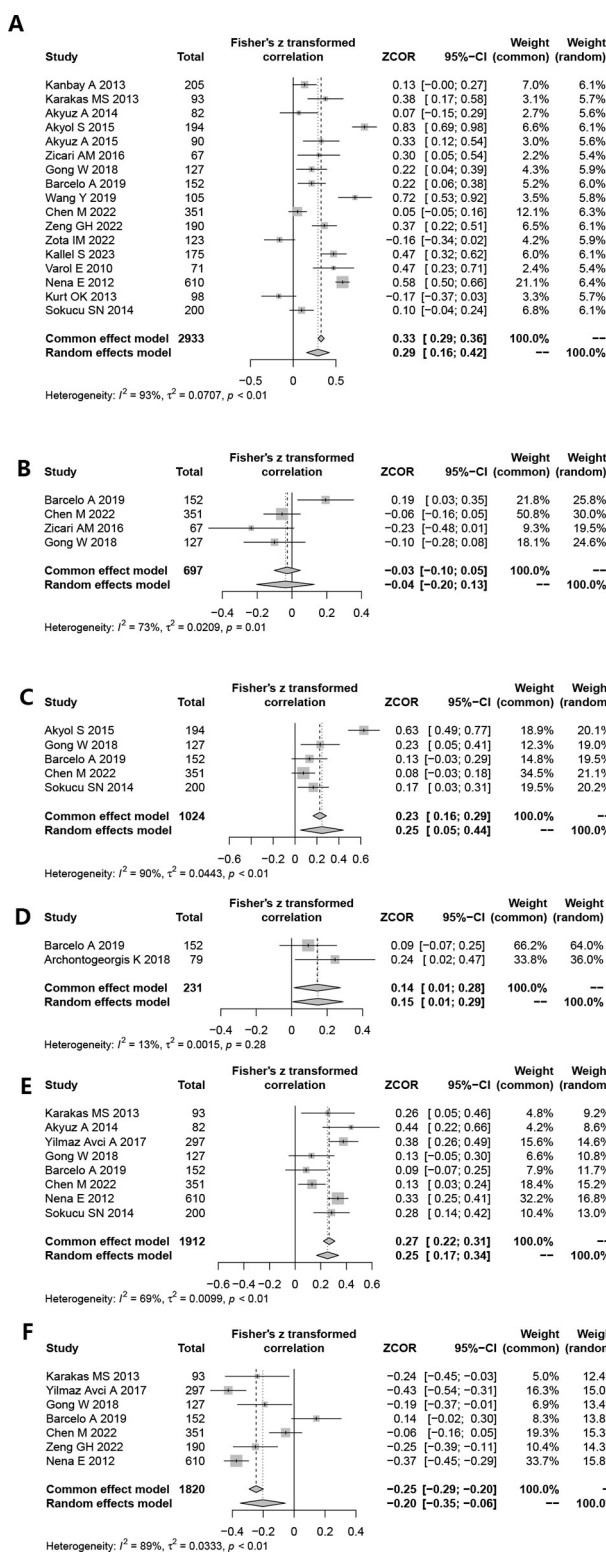

**Fig 5. Funnel plot of effect sizes measured as correlations between MPV and the AHI, Mean SaO₂, ODI, Sleep efficiency, SaO₂<90%, LSaO₂.** A: AHI; B: Mean SaO₂; C: ODI; D: Sleep efficiency; E: SaO₂<90%; F: LSaO₂.

**Correlation between MPV and LSaO$_2$.** In the observed population, seven studies have presented findings regarding the association between MPV and LSaO$_2$. Upon pooling the Pearson correlation coefficients, the meta-analysis findings indicated a weak correlation between MPV and LSaO$_2$ (ZCOR = -0.20, 95% CI = -0.35–0.06, *P* = 0.006) (Fig 5F).

**Publication bias.** Since 17 studies have reported the relationship between MPV and AHI in the included population, a funnel plot was generated to assess publication bias regarding the correlation coefficient between MPV and AHI. The funnel plot manifested no publication bias (S1 Fig). The Egger's test indicated t = -1.05, *P* = 0.310.

## Meta-analysis: The relationship between MPV and the risk of CVDs

Three studies have reported on the relationship between MPV and CVDs risk. Out of these studies, two studies noted that MPV could function as an independent risk factor for CVDs in individuals with OSAHS, while another study pointed out that MPV was an independent risk factor for atrial fibrillation in individuals with OSAHS. A meta-analysis of these three articles was conducted, and the results suggested that MPV was an independent risk factor for CVDs in OSAHS (adjusted OR = 1.72, 95% CI = 1.08–2.73, *P* = 0.022) (Fig 6).

## Discussion

The results of this research showed that MPV values were considerably elevated in individuals with OSAHS compared to the healthy controls, especially in those with severe OSAHS. MPV

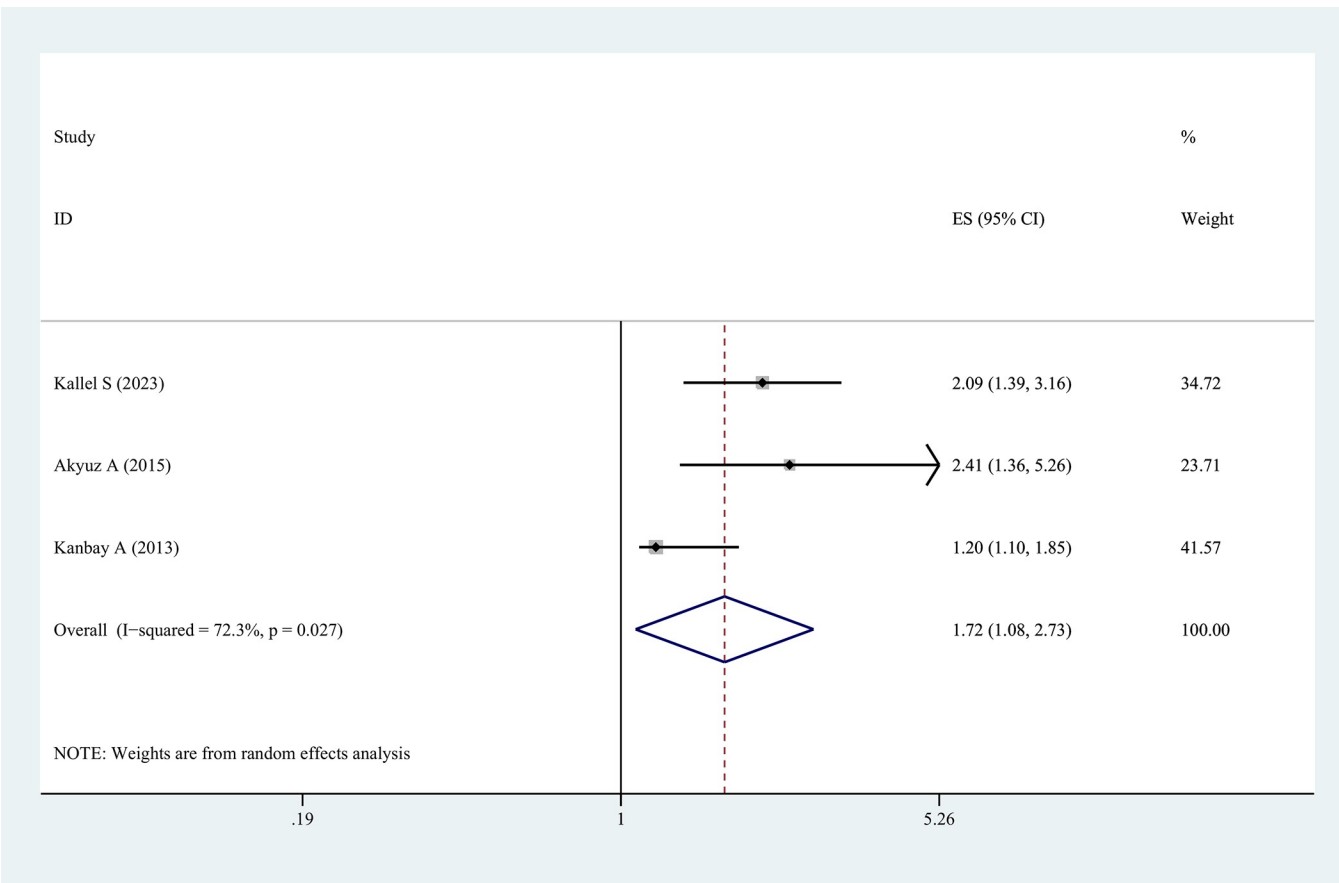

**Fig 6. The forest plot of the relationship between MPV levels and cardiovascular diseases risk in patients with OSAHS.** NOTE: Weights are from random effects analysis.

values were correlated with several sleep monitoring parameters, although the correlation was weak, suggesting an increase in the AHI is associated with a corresponding elevation in MPV values. Furthermore, MPV might be closely associated with the risk of CVDs in individuals with OSAHS. To our knowledge, this is the first comprehensive meta-analysis of the association between MPV and the severity of OSAHS.

MPV is a simple parameter that reflects the increase in platelet activation. It is elevated in various physiological and pathological conditions, including inflammation, hypoxia, and endothelial damage [59]. MPV can be easily measured using automated blood analyzers, making it a feasible marker to assess cardiovascular risk in patients with OSAHS. It was proposed that MPV can serve as an alternative indicator to evaluate inflammation in individuals with OSAHS, quantifying its relationship with AHI, which helps determine the severity of OSAHS. There were proatherogenic molecules, including tumor necrosis factor-a (TNF-a), interleukin-6 (IL-6), oxidized low density lipoprotein cholesterol, correlated to increased resistance to insulin and active macrophages. They affected platelet volume to worsen atherosclerosis and increase cardiovascular disease in patients with OSAHS [60]. Additionally, elevated MPV values can aid in identifying high-risk individuals with OSAHS and alert clinicians to the potential increased risk of CVDs. OSAHS is a chronic condition that can lead to various comorbidities, including respiratory diseases, endocrine dysfunction, and cognitive impairment [61–63]. CVDs are a prominent comorbidity of OSAHS, especially coronary artery disease [64]. MPV is an important indicator for assessing the occurrence of CVDs in individuals with OSAHS.

The association between OSAHS and accompanying changes in MPV values is complex. It may involve three main mechanisms. Firstly, intermittent nocturnal hypoxia and sleep deprivation in OSAHS lead to sympathetic nervous system activation, characterized by increased levels of adrenaline and noradrenaline [65, 66]. Adrenaline and noradrenaline can cause dose-dependent activation of platelets [67]. Narkiewicz *et al.* [68] found that individuals with OSAHS exhibit high sympathetic activity even in a fully awake state, which results in platelet activation and the occurrence of CVDs. Secondly, OSAHS activates oxidative stress and systemic inflammatory reactions, leading to excessive release of interleukin-6. This release further stimulates macrophages, leading to the production of platelets characterized by larger granules. Consequently, this cascade of events culminates in an elevation of MPV values [69]. Platelets with larger volumes contain more dense granules, which can release serotonin and calcium ions. Serotonin can cause platelets to change from a biconcave shape to a rounded shape with pseudopodia, leading to an increase in platelet volume. Additionally, serotonin can foster thrombus formation, resulting in a vicious cycle and increased occurrence of thromboembolic events [42, 70]. Hence, in this study, MPV is considered a risk factor for CVDs in individuals with OSAHS. Thirdly, hypoxemia facilitates the compensatory increase in red blood cells, leading to hemodynamic changes. This alteration subsequently results in platelet thrombus formation, necessitating a substantial quantity of platelets. Compensatory new platelets from bone marrow are considerably larger as compared to mature platelets [71, 72]. These three possible mechanisms explain why MPV in individuals with OSAHS is higher than in healthy individuals.

A study involving 610 participants with polysomnographic records established a notable association between MPV and AHI [34]. The study further highlighted the correlation between MPV and additional sleep monitoring indicators such as the lowest oxygen partial pressure and TS90%. Furthermore, the study observed a consistent pattern wherein MPV values were correlated with various sleep monitoring parameters associated with OSAHS, such as AHI, ODI, and TS90%. Therefore, it is speculated that MPV elevates concomitantly with incremental values of AHI. This phenomenon explains the observed elevation of MPV in patients with

moderate and severe OSAHS in comparison to the healthy controls. Additionally, it is further noted that the increase in MPV values becomes more prominent as the severity of OSAHS escalates. In other subgroup analyses, whether based on ethnicity, mean BMI, or study design type, the MPV values of patients with OSAHS remained higher than those of healthy participants. This finding suggests that ethnicity, mean BMI, and study design type may not be confounding factors for the elevated MPV values in individuals with OSAHS. It is worth noting that in this study, there was no correlation between MPV and mean $SaO_2$, which may be ascribed to the non-parallel relationship between AHI and mean $SaO_2$ [58]. Secondly, there is limited literature on the relationship between MPV and mean $SaO_2$ and the sample size is an important factor that affects the results. For forthcoming studies, it is desirable to include a greater number of original articles to determine the relationship between MPV and mean $SaO_2$.

The population included in this study was mainly adults due to limited research regarding the association between MPV and pediatric OSAHS. The study of Erdim et al. [73] demonstrated no notable correlation between MPV in the blood of obese pediatric patients with OSAHS and the disease. This result may be specific to the study population and subject to selection bias. Onder et al. [74] proposed that the MPV of individuals with OSAHS and adenotonsillar hypertrophy did not necessarily correlate with upper airway obstruction. Nevertheless, Zicari et al. [48] observed 22 children with OSAHS and identified that their MPV values were higher than those of healthy children and those with simple snoring. Their analysis suggested that the heightened platelet volume is a result of the combined effect of systemic inflammatory response. Chang et al. [75] conducted a meta-analysis, and showed that the MPV level is significantly higher in patients with pediatric sleep-disordered breathing compared with the control group. The findings of the present study are consistent with their results. However, their study population was different from the study population included in the present report, as it did not include adult patients with OSAHS. In an addition, OSAHS is only the most common type of sleep-disordered breathing, with the main clinical manifestation of snoring [76]. Furthermore, it is worth mentioning that Chang et al. [75] did not found that the association between pediatric vessel endothelial dysfunction and MPV level in patients with pediatric sleep-disordered breathing. In contrast, the present meta-analysis suggested that MPV was an independent risk factor for CVDs in OSAHS. This might be related to different inclusion criteria and study populations. In conclusion, there is controversy in the literature regarding MPV in pediatric patients with OSAHS; thus, further research is necessary to confirm these findings.

Typically, the heterogeneity between studies in a meta-analysis is related to factors such as the quality of included studies, characteristics of the population, and study design. To investigate the potential sources of heterogeneity, this study conducted subgroup analysis and meta-regression analysis. The meta-regression analysis showed no heterogeneity in terms of ethnicity, mean BMI, disease severity, and study design type. Regardless of the meta-regression results, subgroup analysis was performed based on the aforementioned factors. Unfortunately, no significant sources of heterogeneity were found. Furthermore, a sensitivity analysis was conducted using the one-study removal method, and no individual study was found to cause high heterogeneity. Other factors may contribute to the sources of heterogeneity, such as differences in experimental conditions, variations in the timing of blood sample collection, differences in sample storage methods, environmental smoking status, and lifestyle factors, all of which could be confounding factors influencing MPV values.

## Strengths and limitations

A meta-analysis was performed by Wu et al. [77], showing that there was a positive correlation between the level of MPV and the severity of OSAS. They focused primarily on levels of

Hematological indices among patients with OSAHS, using SMD as effect size. However, considering that most of the studies included in the previous meta-analysis were sourced from Turkey, there may be a certain degree of selection bias. Moreover, the previous meta-analysis did not analyze the relationship between MPV and sleep monitoring indicators such as AHI, oxygen desaturation index (ODI), and $LSaO_2$. When compared to the previous meta-analysis [77], there are several innovations in this meta-analysis. Firstly, this meta-analysis includes studies conducted on both Asian and African populations, which enriches the sources of sample data and increases the generalizability of the results. Secondly, the relationship between MPV and sleep monitoring parameters was analyzed. Quantifying the relationship between MPV and AHI allows outpatient doctors to preliminarily assess the condition of the patients and the severity of hypoxia based on MPV values. Thirdly, previous studies included retrospective studies, but this meta-analysis also includes some cross-sectional studies and case-control studies. This diversification of study designs enhances the reliability of our results. Finally, no significant publication bias was found, indicating that the synthesized results may be reliable.

However, it is also important to acknowledge the specific limitations of this study. Firstly, the majority of the included study participants were adults with OSAHS, thereby warranting more attention to exploring the correlation between MPV and OSAHS in specific populations, such as children with OSAHS. Secondly, due to the lack of sufficient longitudinal cohort studies in this research, a cause-and-effect relationship between OSAHS and MPV could not be firmly established. Additionally, due to the nature of the data, it is difficult to determine the critical value of MPV as an indicator for assessing the severity of OSAHS, and a dose-responsive relationship between MPV and the risk of OSAHS has not been identified. Therefore, further research is needed to clarify these aspects.

## Conclusion

In summary, the MPV in individuals with OSAHS is heightened in comparison to the healthy controls. MPV serves as a relatively simple, cost-effective, and practical indicator of OSAHS severity. Moreover, the increase in MPV may elevate the risk of CVDs associated with OSAHS. However, considering the heterogeneity in the studies included in the research, future investigations seeking to establish the relationship between MPV and OSAHS should employ a unified reference value of MPV, uniform method of testing MPV and generally acknowledged criteria for OSAHS diagnosis.

## Supporting information

**S1 Table. The exact search terms of each database.**
(DOCX)

**S2 Table. The R code.**
(DOCX)

**S1 Fig. Funnel plot of the correlation coefficient between MPV and AHI.**
(TIF)

## Acknowledgments

We thank Bullet Edits Limited for the linguistic editing and proofreading of the manuscript.

## Author Contributions

**Conceptualization:** Jun Zeng, Jie He.

**Data curation:** Jun Zeng, Jie He.

**Formal analysis:** Jie He.

**Funding acquisition:** Jie He.

**Investigation:** Jie He, MeiFeng Chen, Jia Li.

**Methodology:** Jie He, MeiFeng Chen.

**Project administration:** Jie He.

**Resources:** Jie He.

**Software:** Jun Zeng, Jie He, MeiFeng Chen, Jia Li.

**Supervision:** Jun Zeng, Jie He.

**Validation:** Jie He, Jia Li.

**Visualization:** Jie He.

**Writing – original draft:** Jun Zeng, Jie He.

**Writing – review & editing:** Jie He.

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
