## [Decision Letter · Decision Letter 0]

7 Dec 2023

PONE-D-23-26543Association between mean platelet volume and obstructive sleep apnea-hypopnea syndrome: A systemic review and meta-analysisPLOS ONE

Dear Dr. He,

Thank you for submitting your manuscript to PLOS ONE. After careful consideration, we feel that it has merit but does not fully meet PLOS ONE’s publication criteria as it currently stands. Therefore, we invite you to submit a revised version of the manuscript that addresses the points raised during the review process.

We look forward to receiving your revised manuscript.

Kind regards,

Muhammad Tarek Abdel Ghafar, M.D

Academic Editor

PLOS ONE

Journal Requirements:

"Priming Scientific Research Foundation for the Introduced Talents of The First Affiliated Hospital of Chengdu Medical College(CYFY-GQ59)"

5. Please include your tables as part of your main manuscript and remove the individual files. 

Reviewers' comments:

Reviewer's Responses to Questions

**Comments to the Author**

1. Is the manuscript technically sound, and do the data support the conclusions?

Reviewer #1: Yes

Reviewer #2: Yes

Reviewer #3: Yes

2. Has the statistical analysis been performed appropriately and rigorously? 

Reviewer #1: Yes

Reviewer #2: Yes

Reviewer #3: Yes

3. Have the authors made all data underlying the findings in their manuscript fully available?

Reviewer #1: Yes

Reviewer #2: Yes

Reviewer #3: Yes

4. Is the manuscript presented in an intelligible fashion and written in standard English?

Reviewer #1: Yes

Reviewer #2: Yes

Reviewer #3: Yes

5. Review Comments to the Author

Reviewer #1: Zeng et al. have performed a systematic review and meta-analysis study on the relation between MPV and OSAHS. Although the results are interesting, there are some concerns needing to be addressed:

- Abstract: In the methods section, it should be stated that meta-analysis was performed to calculate which measures (SMD, OR). Also, random or fixed effect should be stated.

- Abstract: In the results section of abstract, it is highly suggested to add the total number of patients and controls.

- Introduction is long and it contains unrelated topics in some sections. The authors should keep the introduction short, focusing on main ideas and the rationale and gap of knowledge which led to performing this study.

- Methods: What study designs were included in the systematic review? Was there any limitation (e.g., observational studies only)?

- Methods: Although the definition of OSAHS severity based on AHI is correct, it needs reference.

- Methods: Type of meta-analysis (REML, DL, etc) should be added to statistical analysis section. Moreover, the decision to conduct random-effect or fixed effect meta-analysis should not be made solely based on heterogeneity. As mentioned by https://mentalhealth.bmj.com/content/17/2/64, “If there are important reasons to believe that the relative treatment effect is common in all included studies, then a fixed effect meta-analysis is a reasonable option.” and “When researchers expect that the treatment effects will be similar but not identical then random effects model is the appropriate one to use.” This should be corrected in the methods section.

- Methods: What was the software used for performing the analysis?

- Methods: Although performed, subgroup analysis and meta-regression are not described in statistical analysis section.

- Discussion: The first paragraph of discussion should focus on main findings and the take-home message of the current study.

- In discussion, the comparison of the current study and previous meta-analyses (PMID 30746195 and 32393268) should be made comprehensively.

Reviewer #2: The study titled "Association between mean platelet volume and obstructive sleep apnea-hypopnea syndrome: A systemic review and meta-analysis" conducted by Zeng et al. evaluated the possible diagnostic role of MPV in apnea patients. The study is well-designed. They found promising role for MPV in this disease and can pave the way for future studies. I have some comments:

Introduction:

1- Mention other non-invasive biomarkers for diagnosis and prognosis of apnea using previous systematic reviews and meta-analyses (e.g., HIF-1, Endocan, leptin).

2- Please move the sentence you are mentioning previous meta-analyses on this topic to the discussion. Although it is important to mention that your manuscript is an updated version of previous meta-analyses, the appropriate section to discuss the difference and the added value of your study is the discussion.

Methods:

3- You should mention the PROSPERO code in the first paragraph of the methods, where you mentioned PRISMA guidelines.

4- If possible, add the exact search terms of each database as a supp. table.

5- Mention meta-regression and subgroup analyses in the methods.

Discussion:

6- Add a subheading to your discussion "Strengths and limitations" and mentioned last two paragraphs of the discussion under this subheading.

7- In the conclusions, clearly state how future studies can help confirm your findings.

Reviewer #3: MPV might be a relatively simple, cost-effective, and practical indicator of OSAHS severity.

1.Severity standards for children are different from those for adults

2. Descriptive analysis of differences in MPV between children with OSAHS and controls. The mean platelet volume (fL) ？unit of measurement.

6. PLOS authors have the option to publish the peer review history of their article (what does this mean?). If published, this will include your full peer review and any attached files.

Reviewer #1: No

Reviewer #2: No

Reviewer #3: No

---

## [Author Response · Author response to Decision Letter 0]

20 Dec 2023

Reply to Reviewers： 

Reply: We thank the reviewers’ comments and excellent suggestions. We have revised these mistakes and inaccuracies in the text as you pointed out, rewritten some references and revised some parts of the text. Reply to comments and suggestions of reviewers as follows:

Journal Requirements:

Reply: We have modified the format of the article according to the document.

"Priming Scientific Research Foundation for the Introduced Talents of The First Affiliated Hospital of Chengdu Medical College (CYFY-GQ59)"

Reply: "The funders had no role in study design, data collection and analysis, decision to publish, or preparation of the manuscript." Has been added in the funding statement. Please see line 275-277.

Reply: this meta-analysis was carried out in order to perform secondary research with a higher evidence level, in this way providing hints for basic research and clinical practice. This study did not concern any dataset. Thus, we deleted the content of Data Availability statement.

Reply: this meta-analysis was carried out in order to perform secondary research with a higher evidence level, in this way providing hints for basic research and clinical practice. This study did not concern any dataset. Thus, we deleted the content of Data Availability statement

5. Please include your tables as part of your main manuscript and remove the individual files. 

Reply: we inserted tables in main manuscript. Please see line XXX-XXX.

Reply: we revised Supporting Information and update any in-text citations to match accordingly. Please see line 260-262.

Reviewer #1: Zeng et al. have performed a systematic review and meta-analysis study on the relation between MPV and OSAHS. Although the results are interesting, there are some concerns needing to be addressed:

- Abstract: In the methods section, it should be stated that meta-analysis was performed to calculate which measures (SMD, OR). Also, random or fixed effect should be stated.

Reply: In the methods section, it has been stated that meta-analysis was performed to calculate which measures (SMD, COR, OR). A fixed-effect model was used if the heterogeneity was not significant (I2<50%); otherwise, a random-effect model was applied. We will also combine effect estimates of individual trials using fixed‐effect or random‐effects models. Please see line 29-33.

- Abstract: In the results section of abstract, it is highly suggested to add the total number of patients and controls.

Reply: The study involved 3604 patients and 1165 control individuals. Please see line 36-37.

- Introduction is long and it contains unrelated topics in some sections. The authors should keep the introduction short, focusing on main ideas and the rationale and gap of knowledge which led to performing this study.

Reply: We have removed a portion of the content from the introduction. Please see line 51-89.

- Methods: What study designs were included in the systematic review? Was there any limitation (e.g., observational studies only)?

Reply: We included studies reporting human-only original research (randomized-controlled trials, prospective or retrospective comparatives cohorts, and case-control studies). Please see line 108-109.

- Methods: Although the definition of OSAHS severity based on AHI is correct, it needs reference.

Reply: We have added references. Please see line 115.

- Methods: Type of meta-analysis (REML, DL, etc) should be added to statistical analysis section. Moreover, the decision to conduct random-effect or fixed effect meta-analysis should not be made solely based on heterogeneity. As mentioned by https://mentalhealth.bmj.com/content/17/2/64, “If there are important reasons to believe that the relative treatment effect is common in all included studies, then a fixed effect meta-analysis is a reasonable option.” and “When researchers expect that the treatment effects will be similar but not identical then random effects model is the appropriate one to use.” This should be corrected in the methods section.

Reply: The meta-analysis was conducted using the method of residual maximum likelihood (REML). Fixed or random-effects models were used depending on the between-study heterogeneity (threshold at >50%). Moreover, we will use a fixed effect or random-effect model to estimate the effect across studies, and we will use the fixed effect model if we can reasonably assume the studies are estimating the same effect. Please see line 141-143, line 158-165. 

- Methods: What was the software used for performing the analysis?

Reply: Statistical analysis was carried out by employing STATA 11.0(Stata Corporation, College Station, TX, United States) and R 4.1.3(R Project for Statistical Computing, http://www.r-project.org/). Please see line 141-142.

- Methods: Although performed, subgroup analysis and meta-regression are not described in statistical analysis section.

Reply: We performed the subgroup analyses to determine the efforts of ethnicity, disease severity, BMI and research style on the MPV values between OSAHS patents and control. We also introduced a meta-regression model with ethnicity, disease severity, BMI and research style as independent variables to explore possible heterogeneity. Please see line 161-165.

- Discussion: The first paragraph of discussion should focus on main findings and the take-home message of the current study.

Reply: We have cut out a portion of the discussion content in the first paragraph. Please see line 130-136

- In discussion, the comparison of the current study and previous meta-analyses (PMID 30746195 and 32393268) should be made comprehensively.

Reply: We have cited these two references and carried out a comparison and discussion.

Chang et al conducted a meta-analysis, and showed that the MPV level is significantly higher in patients with pediatric sleep-disordered breathing compared with the control group. The findings of the present study are consistent with their results. However, their study population was different from the study population included in the present report, as it did not include adult patients with OSAHS. In an addition, OSAHS is only the most common type of sleep-disordered breathing, with the main clinical manifestation of snoring. Furthermore, it is worth mentioning that Chang et al [75] did not found that the association between pediatric vessel endothelial dysfunction and MPV level in patients with pediatric sleep-disordered breathing. In contrast, the present meta-analysis suggested that MPV was an independent risk factor for CVDs in OSAHS. This might be related to different inclusion criteria and study populations. Please see line 201-211.

A meta-analysis was performed by Wu et al, showing that there was a positive correlation between the level of MPV and the severity of OSAS. They focused primarily on levels of Hematological indices among patients with OSAHS, using SMD as effect size. However, considering that most of the studies included in the previous meta-analysis were sourced from Turkey, there may be a certain degree of selection bias. Moreover, the previous meta-analysis did not analyze the relationship between MPV and sleep monitoring indicators such as AHI, oxygen desaturation index (ODI), and LSaO2.When compared to the previous meta-analysis, there are several innovations in this meta-analysis. Firstly, this meta-analysis includes studies conducted on both Asian and African populations, which enriches the sources of sample data and increases the generalizability of the results. Secondly, the relationship between MPV and sleep monitoring parameters was analyzed. Quantifying the relationship between MPV and AHI allows outpatient doctors to preliminarily assess the condition of the patients and the severity of hypoxia based on MPV values. Thirdly, previous studies included retrospective studies, but this meta-analysis also includes some cross-sectional studies and case-control studies. This diversification of study designs enhances the reliability of our results.

Please see line 227-241.

Reviewer #2: The study titled "Association between mean platelet volume and obstructive sleep apnea-hypopnea syndrome: A systemic review and meta-analysis" conducted by Zeng et al. evaluated the possible diagnostic role of MPV in apnea patients. The study is well-designed. They found promising role for MPV in this disease and can pave the way for future studies. I have some comments:

Introduction:

1- Mention other non-invasive biomarkers for diagnosis and prognosis of apnea using previous systematic reviews and meta-analyses (e.g., HIF-1, Endocan, leptin).

Reply: we have mentioned other non-invasive biomarkers for diagnosis and prognosis of apnea using previous systematic reviews and meta-analyses. Please see line 66-70.

2- Please move the sentence you are mentioning previous meta-analyses on this topic to the discussion. Although it is important to mention that your manuscript is an updated version of previous meta-analyses, the appropriate section to discuss the difference and the added value of your study is the discussion.

Reply: we have moved the sentence we are mentioning previous meta-analyses on this topic to the discussion. Please see line 85-88. 

Methods:

3- You should mention the PROSPERO code in the first paragraph of the methods, where you mentioned PRISMA guidelines.

Reply: This systematic review protocol has been registered on PROSPERO (https://www.crd.york.ac.uk/PROSPERO/display_record.php?RecordID=451081) with number PROSPERO CRD42023451081. Please see 92-94.

4- If possible, add the exact search terms of each database as a supp. table.

Reply: Please see supplementary table 1.

5- Mention meta-regression and subgroup analyses in the methods.

Reply: We performed the subgroup analyses to determine the efforts of ethnicity, disease severity, BMI and research style on the MPV values between OSAHS patents and control. We also introduced a meta-regression model with ethnicity, disease severity, BMI and research style as independent variables to explore possible heterogeneity. Please see line 161-165.

Discussion:

6- Add a subheading to your discussion "Strengths and limitations" and mentioned last two paragraphs of the discussion under this subheading.

Reply: we added a subheading to our discussion "Strengths and limitations". Please see line 226.

7- In the conclusions, clearly state how future studies can help confirm your findings.

Reply: Future investigations seeking to establish the relationship between MPV and OSAHS should employ a unified reference value of MPV, uniform method of testing MPV and generally acknowledged criteria for OSAHS diagnosis. Please see line 257-259.

Reviewer #3: MPV might be a relatively simple, cost-effective, and practical indicator of OSAHS severity.

1.Severity standards for children are different from those for adults

Reply: We have added references. Please see line 112-115.

2. Descriptive analysis of differences in MPV between children with OSAHS and controls. The mean platelet volume (fL) ？unit of measurement.

Reply: unit of measurement is fL. Line 76-77.

---

## [Decision Letter · Decision Letter 1]

15 Jan 2024

Association between mean platelet volume and obstructive sleep apnea-hypopnea syndrome: A systemic review and meta-analysis

PONE-D-23-26543R1

Dear Dr. He,

We’re pleased to inform you that your manuscript has been judged scientifically suitable for publication and will be formally accepted for publication once it meets all outstanding technical requirements.

Kind regards,

Muhammad Tarek Abdel Ghafar, M.D

Academic Editor

PLOS ONE

Additional Editor Comments (optional):

Reviewers' comments:

Reviewer's Responses to Questions

**Comments to the Author**

1. If the authors have adequately addressed your comments raised in a previous round of review and you feel that this manuscript is now acceptable for publication, you may indicate that here to bypass the “Comments to the Author” section, enter your conflict of interest statement in the “Confidential to Editor” section, and submit your "Accept" recommendation.

Reviewer #1: All comments have been addressed

Reviewer #2: All comments have been addressed

Reviewer #3: All comments have been addressed

2. Is the manuscript technically sound, and do the data support the conclusions?

Reviewer #1: (No Response)

Reviewer #2: (No Response)

Reviewer #3: Yes

3. Has the statistical analysis been performed appropriately and rigorously? 

Reviewer #1: (No Response)

Reviewer #2: (No Response)

Reviewer #3: Yes

4. Have the authors made all data underlying the findings in their manuscript fully available?

Reviewer #1: (No Response)

Reviewer #2: (No Response)

Reviewer #3: Yes

5. Is the manuscript presented in an intelligible fashion and written in standard English?

Reviewer #1: Yes

Reviewer #2: (No Response)

Reviewer #3: Yes

6. Review Comments to the Author

Reviewer #1: The authors have addressed all my comments and no further change is needed. The manuscript is acceptable in its current form.

Reviewer #2: The revised version is suitable for publication in PLOS ONE. The authors have addressed my comments properly.

Reviewer #3: This result is clinically meaningful.It may help doctors complete the assessment of OSAHS more quickly and accurately.The manuscript is in accordance with the Institutional Ethics committee requirements.The content of the article has not been published or submitted for publication elsewhere.

7. PLOS authors have the option to publish the peer review history of their article (what does this mean?). If published, this will include your full peer review and any attached files.

Reviewer #1: No

Reviewer #2: No

Reviewer #3: No

---

## [Editor Report · Acceptance letter]

9 Feb 2024

PONE-D-23-26543R1 

PLOS ONE

Dear Dr. He, 

I'm pleased to inform you that your manuscript has been deemed suitable for publication in PLOS ONE. Congratulations! Your manuscript is now being handed over to our production team.

Kind regards, 

on behalf of

Prof Muhammad Tarek Abdel Ghafar 

Academic Editor

PLOS ONE